# In Situ Growth of Nanosilver on Fabric for Flexible Stretchable Electrodes

**DOI:** 10.3390/ijms232113236

**Published:** 2022-10-31

**Authors:** Qingwei Liao, Yuxiang Yin, Jingxin Zhang, Wei Si, Wei Hou, Lei Qin

**Affiliations:** 1Key Laboratory of Sensors, Beijing Information Science and Technology University, Beijing 100192, China; 2School of Instrument Science and Opto-Electronics Engineering, Beijing Information Science and Technology University, Beijing 100192, China; 3Key Laboratory of Modern Measurement and Control Technology, Ministry of Education, Beijing Information Science and Technology University, Beijing 100192, China; 4Key Laboratory of Photoelectric Testing Technology, Beijing Information Science and Technology University, Beijing 100192, China

**Keywords:** flexible, conductive fabric, in situ growth

## Abstract

Flexible sensing can disruptively change the physical form of traditional electronic devices to achieve flexibility in information acquisition, processing, transmission, display, and even energy, and it is a core technology for a new generation of the industrial internet. Fabric is naturally flexible and stretchable, and its knitted ability makes it flexibility and stretchability even more adjustable. However, fabric needs to be electrically conductive to be used for flexible sensing, which allows it to carry a variety of circuits. The dip-coating technique is a common method for preparing conductive fabrics, which are made conductive by attaching conductive fillers to the fabrics. However, the adhesion of the conductive fillers on the surface of such conductive fabrics is weak, and the conductive property will decay rapidly because the conductive filler falls off after repeated stretching, limiting the lifespan of flexible electronic devices based on conductive fabric. We chose multifunctional nanosilver as a conductive filler, and we increased the adhesion of nanosilver to fabric fiber by making nanosilver grow in situ and cover the fiber, so as to obtain conductive fabric with good conductivity. This conductive fabric has a minimum square resistance of 9 Ω/sq and has better electrical conductivity and more stable electrical properties than the conductive fabric prepared using the dip-coating process, and its square resistance did not increase significantlyafter 60 stretches.

## 1. Introduction

The industrial internet is undergoing a transformation. With the new generation of information and communication network technology, the manufacturing industry will complete the transformation of digitization, networking, and intelligence, of which the key technology is perception. Moreover, flexible perception is the key technical support for multidimensional space perception. Flexible sensing changes the rigid physical form of traditional information devices and makes the acquisition, transmission, processing, and display of information more flexible to realize the efficient integration of information and people. Currently, flexible sensing technology has been widely applied in many fields, such as medical biology, human–computer interaction, environmental monitoring, and wearable technology [1,2,3,4,5]. As a key material in flexible sensing technology, flexible conductive materials are receiving more and more attention and research [6,7]. Conductive fabric is a special kind of flexible conductive material, which not only has good stretchability and breathability of the fabric, but also has high conductivity, so it has great potential for application in the field of flexible sensing, especially in the field of wearable technology [8]. Highly conductive, comfortable, and stretchable fabrics have become a research hotspot in the field of flexible sensing [9].

There are usually two methods of preparing conductive fabric, the first of which is to embed metal wires into the fabric fibers through a weaving process [10,11,12]. Although fabric with mixed metal wires has good conductivity, the metal wires increase the stiffness and weight of the fabric and reduce the tensile properties of the fabric, which seriously affects wearing comfort [13], and the metal wires in the fabric fibers tend to break during the bending and stretching process, resulting in reduced conductivity. The second method is through electrochemical deposition, dip-coating, and other techniques to deposit conductive fillers on ordinary fabric [14,15,16,17]. The method allows conductive fabric with corresponding properties to be prepared according to the characteristics of conductive fillers. Common conductive fillers include graphene, carbon nanotubes, conductive polymers, and metal nanomaterials [18,19,20,21]. Selecting different fillers enables the production of conductive fabric with electromagnetic shielding, antibacterial, and flame-retardant properties [22,23,24,25]. The method of depositing nano-conductive layers on the fabric does not affect the softness and comfort of the fabric on a macroscopic level because the conductive filler is at the nanoscale. However, the conductivity of this kind of conductive fabric is low, due to the weak adhesion of the conductive filler on the fabric surface. After a certain number of stretches, the conductive filler on the fabric surface fell off, resulting in a decrease in the conductivity of the fabric. In situ growth is a method of preparing composites that uses material as a matrix on which another functional body is grafted, polymerized, single-loaded, and deposited [26,27,28]. The composite material obtained from in situ growth has the advantages of both and is more strongly bonded than other methods. Due to the simplicity of the process and the ease of combining different functional materials, the in situ growth method has been widely used for the preparation of various multifunctional composites [29,30,31]. Montes [32] et al. solved the problem of weak adsorption of Ag nanoparticles on the surface of composite fabric by in situ growth of Ag nanoparticles on different types of textile fibers so that Ag nanoparticles uniformly covered the surface and interior of the fabric. Kim et al. [33] prepared a stretchable fiber strain sensor based on an in situ growth process by using mild and harmless sodium ascorbate as a reducing agent, with a low square resistance of 0.9 Ω/cm and high stability over repeated tensile release cycles (5000 cycles). Although the above conductive fabric prepared by in situ growth further strengthens the bond between the nanosilver and the fabric, the nanosilver particles on the surface of the fibers are scattered in various places and do not form a connection with each other. The increase of the distance between the Ag nanoparticles leads to the fracture of the original conductive pathway, which affects the electrical properties of the conductive fabric under a tensile state.

To solve these problems, we plan to grow Ag nanoplates directly on the fabric surface by in situ growth method and interconnect them to enhance the connection between Ag nanoplates. The Ag seeds were formed on the surface and inside the fabric fibers during the pretreatment phase. The growth of the Ag seeds into Ag nanoparticles was promoted by continuous reduction of Ag^+^ ions, and the Ag nanoparticles continued to grow laterally in the presence of the capping agent citric acid to form Ag nanoplates with the (111) facets as the main facets. The conductive fabric was tested with a minimum square resistance of 9 Ω/sq. Compared with the conductive fabric prepared by using the dip-coating process, the conductive fabric prepared by in situ growth technique not only has better electrical conductivity but also has stronger tensile properties. After 50 stretches at a 20% stretching rate, the square resistance of the dip-coated conductive fabric increased by approximately 300%, while the square resistance of the in situ growth conductive fabric was approximately 30%. At 100% stretching, the square resistance of the dip-coated conductive fabric increased from 16 Ω/sq to 55 Ω/sq, while the square resistance of the in situ growth conductive fabric increased from 9 Ω/sq to 12 Ω/sq after stretching.

## 2. Results and Discussions

Mirkin et al. [34] successfully synthesized Ag nanoplates for the first time, using a photo-induced chemical method. This method used sodium citrate as the reducing agent and Bis(*p*-sulfonatophenyl) phenylphosphine dihydrate dipotassium salt solution (BSPP) as the stabilizer. Citric acid is considered to be a key reaction component in the synthesis of Ag nanoplates [35]. Kilin et al. [36] suggested that citric acid, as a capping ligand, preferentially combined with (111) facets and prevented the continued growth of seed on this plane during the formation of Ag nanoplates. Since the symmetry of citrate and seed facets (111) overlap, four Ag-O bonds could be formed, resulting in binding energy greater than that of (100) facets. PVP is a common synthetic material in the process of Ag nanoplates synthesis, which is usually used as a stabilizer to reduce the reduction rate of silver atoms. This is due to the strong binding force between PVP and Ag seed (100), and a small amount of PVP can also increase the thickness of Ag nanoplates [37,38,39]. Washio et al. [40] synthesized Ag nanoplates by using only PVP as a reducing and stabilizing agent under hydrothermal reaction conditions. Since then, researchers have successfully prepared Ag nanoplates by silver mirror method and polyol method, but these methods are difficult to meet the need for a controlled reaction process. The seed-mediated method is an easy way to control the Ag nanoplates’ growth process. It can be used to obtain products with different morphologies by controlling the reaction conditions, such as the use of different capping agents, reducing agents, and stabilizers. The method is divided into two stages: the first stage is nucleation, and the second stage is growth and formation. Wang et al. [41] synthesized Ag nanoplates with diameters of 500 nm–2.2 μm, using NaBH_4_ as the reducing agent to form crystal seeds and ascorbic acid as the reducing agent for electrolyzing growth, but as this method is more complex, we simplified it on top of this by using PVP as the stabilizer and reducing agent in the first step. The seeds were formed by reducing Ag^+^ in the solution, and in the second step, ascorbic acid was used as the reducing agent, which reduced Ag^+^ with the following reactions:(1)$$C_{6}H_{8}O_{6}+2{\mathrm{Ag}}^{+}+2{\mathrm{OH}}^{-}\;\to {\;C}_{6}H_{6}O_{6}+2\mathrm{Ag}\;+2H_{2}O$$

To obtain the desired morphology, the reduction reaction needs to proceed slowly and steadily. Ascorbic acid is more suitable for the seeds’ growth than NaBH_4_ [42] due to its weaker reducing properties. During the growth process, the growth direction of the seeds was regulated by citric acid to induce the growth of the crystalline seeds into Ag nanoplates.

The process of in situ growth of Ag nanoplates on fabric is shown in Figure 1. As a pretreatment, the fabric was soaked in a mixture of PVP and ethanol. PVP can be used as a reducing agent to reduce Ag^+^ and provide Ag^0^ and can also be complexed with Ag^+^ as a stabilizer to reduce the reaction speed and prevent the rapid aggregation of Ag^0^. Newly formed Ag^0^ nanoplates are constantly being reduced under the action of PVP, and when Ag^0^ nanoplates reach a certain concentration in solution, they will collide with each other and aggregate into nuclei [43], eventually forming Ag seeds. The seeds distributed evenly on the surface and inside the fabric require a steady supply of Ag^0^ to continue to grow. Ascorbic acid was used to reduce Ag^+^ to provide the Ag^0^ for seed growth, and citric acid was used as an assistant to limit seed growth at facets (111). Ag seeds are induced to grow in Ag nanoplates and eventually wrap fibers.

Common fabrics include polyester, natural cotton, and linen. Polyester has excellent tensile property and chemical stability, while natural cotton and linen has good air permeability and comfort. They are often used as bases for flexible conductive materials. Polyester has many braid structures; in order to find a suitable fabric substrate, four polyester fabrics with different structures were selected for comparison: large-pore polyester fabric, small-pore polyester fabric, running-stitch polyester fabric, and ribbed polyester fabric. As shown in Figure 2, large-pore polyester fabric can be stretched from 2 cm to 4 cm, but the fibers are less and generally loose. The small-pore polyester fabric can be stretched from 3.5 cm to 5.5 cm, and the tensile property and fiber density are improved. The running-stitch polyester fabric can be stretched from 3.5 cm to 7.5 cm, with excellent tensile properties and high fiber density. The ribbed polyester fabric can be stretched from 1.5 cm to 4 cm, with the highest fiber density. Compared with other fabric structures, the ribbed structure has a smaller spacing between the fibers and a close connection between the fibers, which is conducive to the growth of Ag nanoplates. After the conductive filler is covered, it is easier to form an efficient conductive path. The stretchability of ribbed polyester fabric is weaker than that of running-stitch polyester fabric but stronger than the other two kinds of fabric. In fact, fabric with good tensile properties tends to permanently damage conductive pathways under extreme tensile conditions. After shrinkage, the conductive path originally formed by the interconnections of Ag nanoplates is disconnected, resulting in an irreversible increase in the overall resistance.

We grow Ag nanoparticles in situ on these four fabrics and find suitable fabric substrates for growing Ag nanoparticles by observing the coverage of Ag nanoparticles on the surface of four different fabric fibers. As shown in Figure 3a–d, no obvious Ag nanoparticles were observed on the fiber surface of polyester fabrics with large pores and small pores. As shown in Figure 3e–h, there are obvious Ag nanoparticles on the fiber surface of both running-stitch fabric and ribbed polyester fabric, and the fiber surface of ribbed polyester fabric is covered with the most Ag nanoparticles. After comprehensive consideration, we chose ribbed polyester fabric and natural cotton and linen as substrates and attempted to grow Ag nanoplates in situ on the fibers.

After the in situ growth process, the morphology of natural cotton and flax has changed greatly (Figure 4a,b). Due to the poor chemical stability of natural cotton and flax, it disintegrates after soaking in the reaction solution for a long time and being heated. Figure 4d,e show the conductive fabric after the in situ growth process. The treated conductive fabric is black and gray, as a whole, due to the cover of Ag nanoplates on the surface, but the fabric structure and shape do not change greatly. Figure 4c,f are SEM images of conductive cotton and linen fibers and conductive fabric fibers, respectively. The surface of conductive cotton fiber is mostly Ag nanoparticles, and no Ag nanoplate is formed to cover the fiber. The surface of conductive fabric fiber is coated with Ag nanoplates. According to the SEM image of conductive fabric fiber, the thickness of silver nanosheet is about 8 nm–10 nm. As shown in Figure 4g,h, the stretch range of the treated conductive fabric is 1.5 cm to 3 cm. The initial length and tensile length of the conductive fabric are reduced compared with the fabric before treatment. This is because the bond between fiber molecules is broken when the polyester fabric is soaked in solution and heated at a high temperature during the preparation process, leading to fiber relaxation and contraction. However, the ultimate tensile rate of the fabric before and after treatment was around 100%, indicating that the in situ growth process did not affect the stretchability of the fabric after treatment.

The conductive fabrics prepared by dip-coating process were obtained by soaking in Ag nanowires’ dispersion liquid. Figure 5a,b are SEM images of the fiber surface of conductive fabric by the dip-coating process. It can be seen that nanosilver wires of uniform diameter cover and wrap around the fiber surface. In contrast, the in situ growth process was used to grow Ag nanoplates on natural cotton and linen fabrics and ribbed polyester fabrics. Figure 5c,d show the SEM image of conductive fabric fiber based on natural cotton and flax. The diameter of silver nanoparticles on the surface of natural cotton and linen fabric is about 50–150 nm, showing a stacked distribution without forming Ag nanoparticles. This is due to the production of active hydroxyl groups in the cellulose molecular chains of natural cotton and linen immersed in acid–base solution under continuous heating and stirring, and the crosslinking reaction with citric acid to form ester bonds, consuming a large amount of citric acid. In the absence of sufficient citric acid control, the final Ag nanoparticles freely aggregate to form irregularly shaped Ag nanoparticles. Figure 5e,f show the SEM images of conductive fabric treated by the in situ growth process. It can be seen that the Ag nanoplates grow around the shape of the fiber and finally form the coating.

As shown in Figure 6, when the fabric is stretched, the contact area of the fiber increases with the increase of the stretching degree, the originally interconnected Ag nanowires separate, and the conductive pathway breaks off and falls off. For conductive fabrics prepared by dip-coating process, the binding force between Ag nanowires on the fiber surface and the fiber is mainly derived from Van der Waals forces [44]. The polyester fibers in an acidic and alkaline environment during the preparation of the in situ growth process are prone to hydrolysis, thus leading to the breakage of macromolecular chains and the formation of hydroxyl and carboxyl groups on the surface of the fibers with the free silver ions in solution forming ionic bonding, making the connection between the Ag nanoplates and the fibers tighter and harder to break. In addition, the Ag nanowires on the surface of the conductive fabric prepared by the dip-coating process cover the surface of the fiber but not the interior of the fiber. The in situ growth process gives the fiber Ag nanoplates that cover both the surface and the interior of the fiber by means of a more thorough surface modification, which means better electrical conductivity. In a word, the in situ growth process can improve the binding force between Ag nanoplates and fabric fibers, increase the conductive path between Ag nanoplates, and improve the conductive properties and stability of conductive fabric.

Figure 7a shows the UV–Vis absorption spectrum of the prepared Ag nanowires; there are two relatively sharp absorbance peaks at 350 nm and 380 nm. The peak at 380 nm can be assigned to the transverse local surface plasmon resonance (LSPR) band of long Ag nanowires, while the shoulder peak at 350 nm can be assigned to the longitudinal LSPR band of the bulk Ag nanowires. The UV–Vis spectra demonstrated the bulkiness of the Ag nanowires’ preparation in the product via longitudinal LSPR peak. The high proportion of Ag nanowires synthesis in the product resulted in an intense and sharp LSPR peak in response to UV–visible wavelength. As shown in Figure 7b, the XRD pattern of conductive fiber in the in situ growth process shows a section of amorphous peak at a low angle, which is the diffraction peak of polyester fiber, and there are obvious diffraction peaks at 2θ = 38.3°, 44.4°, and 65° corresponding to the (111), (200), and (220) crystallographic planes of pure face-centered cubic (FCC) phase Ag (JCPDS card no. 4-0783). The XRD patterns of the conductive fibers grown in situ show distinctive diffraction peaks at 2θ of 38.3°, 44.4°, and 65°, indicating that the Ag nanoplates have a high crystallinity, and all the diffraction peaks can be well-indexed to FCC structure (JCPDS card No. 4-0783).

The conductive fabric from the two different processes was stretched to 120% of its original length and then released, and this action was repeated 0–100 times to observe the change in square resistance. The four-probe method is used for measurement. For the instrument equipped with a special four-probe head with spring, the distance of the four probe heads is equal, and the four leads are connected to the square-resistance tester. When the probe is inserted into the fabric, the value of the square-resistance tester is read. For the same fabric, after 10 measurements, the maximum and minimum values were removed, and the mean tail-cut value was calculated as the final result. Figure 7c shows the electrical properties of the conductive fabric after repeated stretching many times. It can be seen from the figure that the square resistance of the dip-coated conductive fabric increased by 10% after repeated stretching and releasing 20 times at a tensile rate of 20%. Then the square resistance increased faster with the increase in the number of stretching. After repeated stretching 60 times, the structure of the dip-coated conductive fabric was damaged and could not be restored to the initial state, and its resistance rose sharply. Under the same test conditions, after 60 tensile releases, the square resistance of the in situ growth conductive fabric was only slightly increased by 30%, and it still had good electrical conductivity. Figure 7d shows the resistance changes of the two kinds of conductive fabric under different tensile degrees. It can be seen that the square-resistance change of dip-coated conductive fabric was relatively small when the tensile rate was 0–20%. The square resistance increased with the increase of the tensile degree and increased sharply when the tensile rate exceeded 60%. This is mainly because the Ag nanowires on the surface of the conductive fabric fiber prepared by the dip-coating process break and fall off with the extension of the fabric, leading to the disconnection of part of the conductive pathway, which affects the electrical conductivity of the fabric. The in situ growth conductive fabric can be bonded more closely with the Ag nanoplates through the interaction between ions [45], and the connection between each other is tighter, so the conductive path on the surface is not easy to break when the fabric is stretched. Figure 7e,f respectively show the relative variation of square resistance of conductive fabrics prepared by two different processes with respect to the number of draws and the drawing rate. It can be seen that the variation of square resistance of conductive fabric prepared by dip-coating process increases with the increase of drawing times and drawing rates. However, the growth rate of square resistance of conductive fabric prepared by in situ growth process did not increase gradually and did not change significantly after 60 times of stretching and within 100% tensile rate.

## 3. Materials and Methods

Propylene glycol (150 mL), ethylene glycol (50 mL), and Polyvinylpyrrolidone (PVP) (6.7 g) were mixed in a beaker labeled as Solution A. Propylene glycol (60 mL), ethylene glycol (20 mL), and AgNO_3_ were mixed and labeled as Solution B. Solution A was treated by magnetic stirring until the solid was completely dissolved and the solution turned pale yellow. Then KBr (0.075 g) and AgCl (0.255 g) were added into Solution A and left at 120 °C for 1.5 h, under stirring and heating, and Solution B was added into Solution A through a constant flow pump. After calefaction, the solution was cooled to room temperature naturally. Then add acetone and let stand for 4 h. Ag nanowires (AgNWs) were obtained after centrifugation and washing.

The fabric was cut into small rectangular pieces of 0.5 cm × 1 cm and cleaned with acetone, methanol, ethanol, and deionized water by ultrasonic cleaning for 15 min, respectively. Then it was put into an electric blast dryer at 60 °C for 20 min to obtain clean fabric pieces. The fabric was soaked in 10 mg/mL of Ag nanowires’ dispersion, and after soaking for some time, it was removed with clean tweezers and put into an electric blast dryer at 75 °C.

The fabric was cut into regular 0.5 cm × 1 cm rectangular pieces. Ethanol (8 mL), ammonia (6 mL), and 1.2 g of silver nitrate were mixed and labeled as Solution A. 1 g PVPmv = 360,000 was dispersed in 40 mL ethanol and labeled as Solution B. Ethanol (60 mL), AgNO_3_ (36 mg), and citric (36 mg) were mixed and labeled as Solution C. The fabric pieces were first soaked in Solution A for 6 h, and then the soaked fabric pieces were put into Solution B and stirred at 75 °C until the color of the solution did not change. Next, the fabric pieces were put into Solution C, and 12 mg/mL ascorbic acid was dropped at a rate of 1.4 mL/min and stirred for 3 h. Finally, the fabric samples were washed with deionized water and ethanol three times and dried at 70 °C. Figure 8 shows the process diagram of Ag nanoparticles in situ growing in the fabric.

## 4. Conclusions

We have developed a simple method of in situ growing nanosilver on fabric to obtain conductive fabric with good stretchability. The conductive fabric was measured by using a four-probe square-resistance tester in comparison with the conductive fabric prepared by the dip-coating process. The square resistance of the dip-coated fabric was as low as 17 Ω/sq, and the square resistance of the in situ growth fabric was as low as 9 Ω/sq. At a 20% tensile rate, the square resistance of the dip-coated fabric increased slightly at 30 repetitions of tensile release, while the square resistance of the in situ growth fabric did not change significantly. After more than 30 repeated stretches, the square resistance of the dip-coated fabric rose sharply, reaching 70 Ω/sq after 60 stretches, an increase of approximately 300%, while the square resistance of the in situ growth fabric remained stable after 60 stretches, reaching 12 Ω/sq after 60 stretches, an increase of approximately 30%. The square resistance of the dip-coated conductive fabric reached 55 Ω/sq at a 100% tensile rate, while the square resistance of the in situ growth conductive fabric only slightly increased to 12 Ω/sq. The in situ growth conductive fabric had a larger conductive area and a stronger bond between the nanosilver and the fabric because the fiber surface was wrapped by Ag nanoplates. Compared with the dip-coated fabric, the in situ growth fabric could have better electrical conductivity and stretchability.

## Figures and Tables

**Figure 1 ijms-23-13236-f001:**
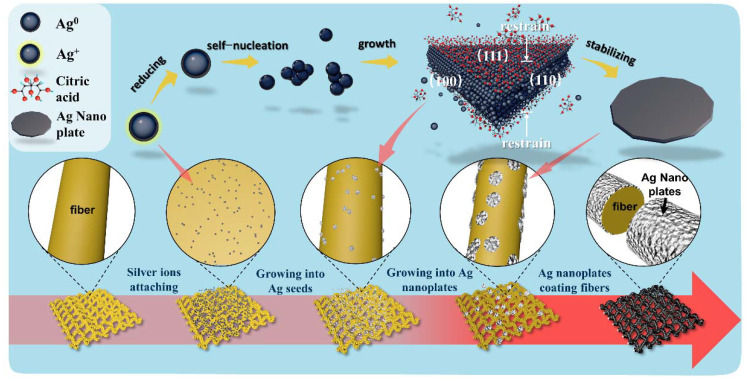
Growth mechanism of Ag nanoplates and growth process of Ag nanoplates on fabric.

**Figure 2 ijms-23-13236-f002:**
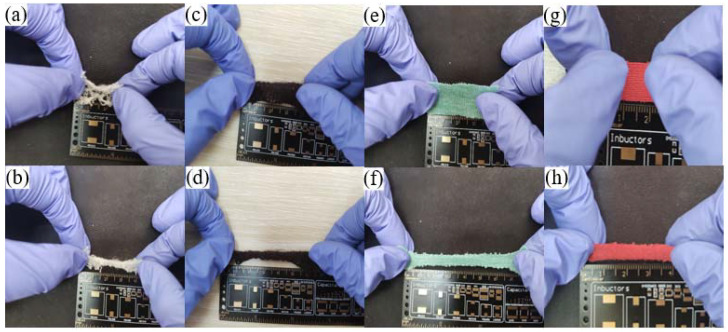
(**a**) Large-pore polyester fabric before stretching. (**b**) Large-pore polyester fabric after stretching. (**c**) Small-pore polyester fabric before stretching. (**d**) Small-pore polyester fabric after stretching. (**e**) Running-stitch polyester fabric before stretching. (**f**) Running-stitch polyester fabric after stretching. (**g**) Ribbed polyester fabric before stretching. (**h**) Ribbed polyester fabric after stretching.

**Figure 3 ijms-23-13236-f003:**
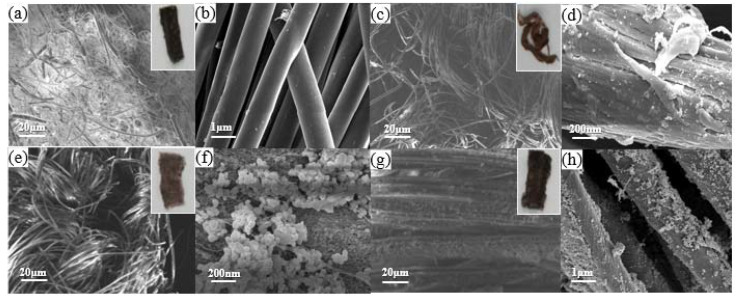
(**a**) Scanning Electron Microscope (SEM) image of conductive fabric prepared by large-pore polyester fabric, and the inset is the physical picture of conductive fabric prepared by large-pore polyester fabric. (**b**) Magnification of large-pore polyester fabric surface. (**c**) SEM image of conductive fabric prepared by small-pore polyester fabric; the inset is the physical image of conductive fabric prepared by small-pore polyester fabric. (**d**) Magnification of small-pore polyester fabric surface. (**e**) SEM image of conductive fabric prepared by running-stitch polyester fabric; the inset is the physical picture of conductive fabric prepared by running-stitch polyester fabric. (**f**) Magnification of running-stitch polyester fabric surface. (**g**) SEM image of conductive fabric prepared by ribbed polyester fabric, and the inset is the physical image of conductive fabric prepared by ribbed polyester fabric. (**h**) Magnification of ribbed polyester fabric surface.

**Figure 4 ijms-23-13236-f004:**
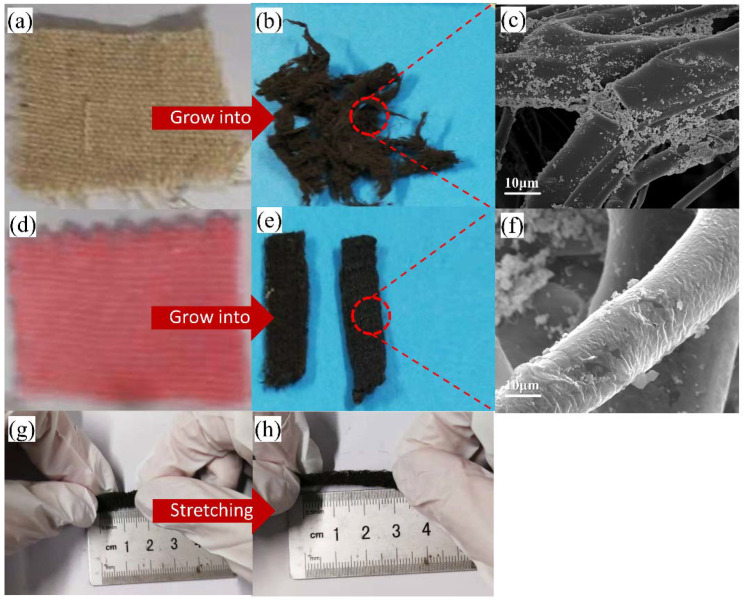
(**a**) The physical image of natural cotton and linen. (**b**) Conductive cotton and linen. (**c**) SEM image of conductive cotton and linen fiber. (**d**) The physical image of ribbed polyester fabric. (**e**) Conductive fabrics. (**f**) SEM image of conductive fabric fiber. (**g**) Conductive fabric before stretching. (**h**) The conductive fabric after stretching.

**Figure 5 ijms-23-13236-f005:**
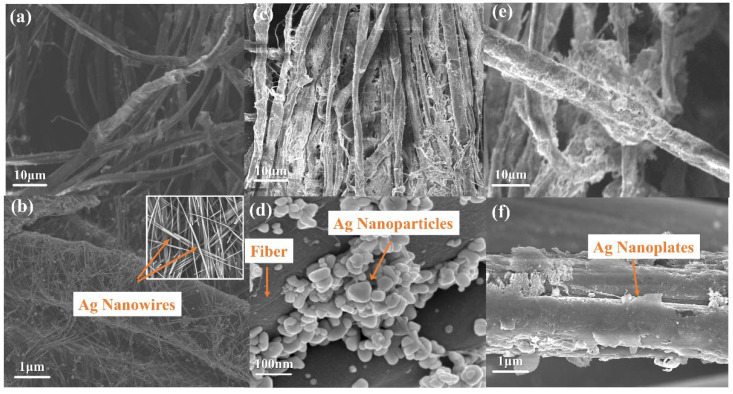
(**a**,**b**) SEM image of conductive fabric fiber prepared by dip-coating process. (**c**,**d**) SEM images of natural cotton and linen fabric fibers processed by in situ growth process. (**e**,**f**) SEM images of conductive fabric fibers prepared by in situ growth process.

**Figure 6 ijms-23-13236-f006:**
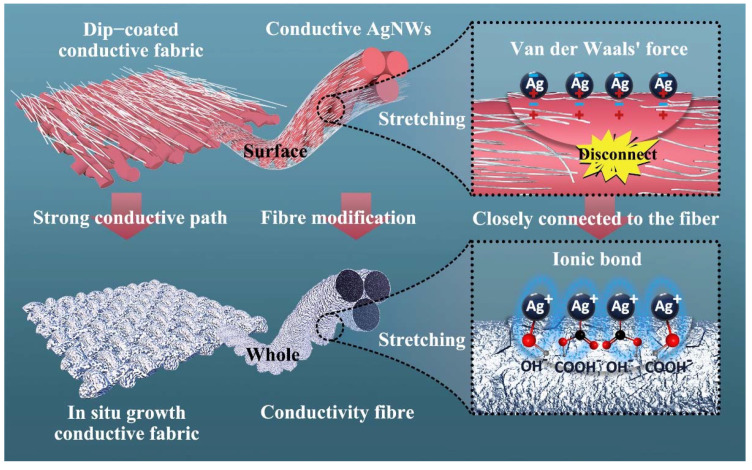
Tensile and electrical conductivity of conductive fabric by different processes.

**Figure 7 ijms-23-13236-f007:**
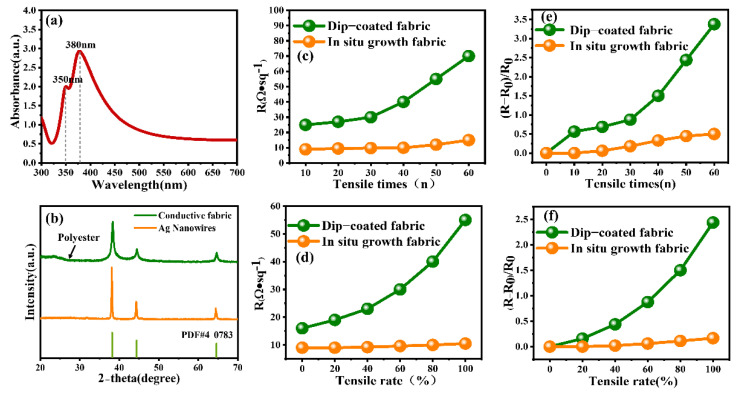
(**a**) UV–Vis image of Ag nanowires. (**b**) X-ray diffraction (XRD) patterns of in situ growth conductive fabric and Ag nanowires. (**c**) The square resistance of two kinds of conductive fabric changes after stretching up to 60 times. (**d**) The square resistance of two kinds of conductive fabric changes from tensile rate to 100%. (**e**) Normalized relationship between square-resistance change and tensile times. (**f**) Normalized relationship between square-resistance change and tensile rate.

**Figure 8 ijms-23-13236-f008:**
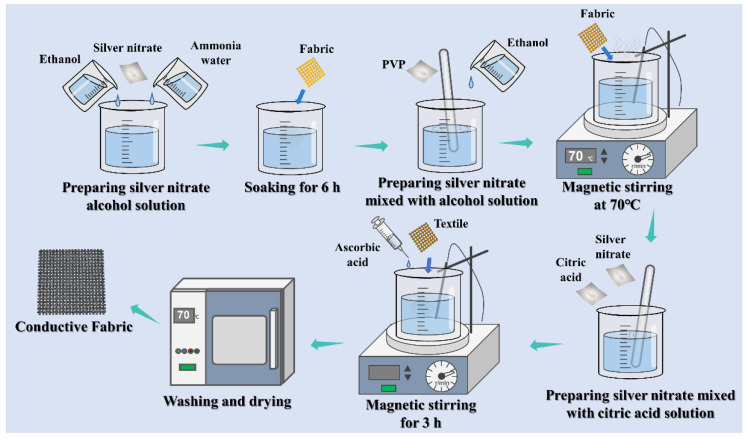
The in situ growth process for the preparation of conductive fabric.

## Data Availability

Not applicable.

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
