# Peer review of "In Situ Growth of Nanosilver on Fabric for Flexible Stretchable Electrodes"

_ijms, 2022, doi:10.3390/ijms232113236_

Round 1

Reviewer 1 Report

The manuscript presents the fabrication of conductive fabric based on nanosilver by the in-situ growth method. This topic is interesting for publication in International Journal of Molecular Sciences. However, the authors should revise the article according to the following, before I can make a positive recommendation regarding publication.

1.     Did the authors perform any optimization for the in-situ growth method such as temperature and solution concentration?

2.     What is the thickness of the Ag nano film in-situ growth on fabric and linen? Is the thickness controllable? The detailed information of the Ag nano film should be given in the text.

3.     In Fig. 5, Ag nanoparticles and Ag nanoplate are grown on the natural cotton and linen fabrics and ribbed polyester fabrics by the same in-situ growth method, what is the reason?

4.     What is the meaning of tensile rate (0%~100%) in Fig. 7 (d) and (f)?

Author Response

First, we would like to express our thanks to the reviewers for their instructive comments concerning our manuscript entitled ‘In-situ growth of nanosilver on fabric for flexible stretchable electrodes’ (ijms-1976254). We have studied the comments carefully, and then carefully revised the whole manuscript. The revisions/ explanation corresponding to the comment are shown in the following (Reviewers' comments are in italic font):

  1. Did the authors perform any optimization for the in-situ growth method such as temperature and solution concentration?

A: Yes, we have tried different temperature and solution concentration, but most of the in-situ growth conditions are not ideal, and finally the successful preparation of in situ growth conductive fabric is described in the article.

  1. What is the thickness of the Ag nano film in-situ growth on fabric and linen? Is the thickness controllable? The detailed information of the Ag nano film should be given in the text.

A:The thickness of the Ag nanoplates grown in-situ on the fabric is about 8-10nm. By adjusting the concentration of PVP in the solution, the thickness of Ag nanoparticles can be adjusted in a certain range. According to the reviewer's comments, we have added a detailed description of the Ag nanoplates in the article:

The surface of conductive fabric fiber is covered with Ag nanoplates. According to the SEM image of conductive fabric fiber, the thickness of Ag nanoplates is about 8 nm-10 nm.

  1. In Fig. 5, Ag nanoparticles and Ag nanoplate are grown on the natural cotton and linen fabrics and ribbed polyester fabrics by the same in-situ growth method, what is the reason?

A:We tried to grow Ag nanoplates on natural cotton and linen and polyester at the same time. In the same way, Ag nanoplates could not be formed on natural cotton and linen. After discussion, it is concluded that due to the poor chemical stability of natural cotton and linen in the high temperature environment of in-situ growth process, molecular chain breakage and binding with citric acid affect the growth of silver nanosheets. The details are as follows:

This is due to the production of active hydroxyl groups in the cellulose molecular chains of natural cotton and linen immersed in acid-base solution under continuous heating and stirring, and the cross-linking reaction with citric acid to form ester bonds, consuming a large amount of citric acid. In the absence of sufficient citric acid control, the final Ag nanoparticles freely aggregate to form irregular shape Ag nanoparticles.

  1. What is the meaning of tensile rate (0%~100%) in Fig. 7 (d) and (f)?

A: The stretch rate of the fabric is the percentage of the length of the stretched fabric to the original fabric along the stretch direction. The stretch rate of 0%-100% refers to the stretch of the fabric from its original length to 200% of its original length.

In this revising chance, we revised the manuscript over and over again, besides all revisions shown above, there are other modifications, which are shown as below:

Page 2. Line 50: “have” has been revised as “has”.

Page 2. Line 65: “of” has been revised as “in”.

Page 2. Line 66: “uses a material” has been revised as “uses material”.

Page 2. Line 75: “a reducing agent” has been revised as “reducing agent”.

Page 2. Line 82: “under tensile state” has been revised as “under a tensile state”.

Page 2. Line 92: “have stronger tensile proper-ties” has been revised as “has stronger tensile properties”.

Page 3. Line 105: “the solution A” has been revised as “solution A”.

Page 3. Line 95: “PVP” has been revised as “Polyvinylpyrrolidone(PVP)”.

Page 3. Line 102: “AgNWs” has been revised as “Ag nanowires(AgNWs)”.

Page 3. Line 95: “SEM” has been revised as “Scanning Electron Microscope (SEM)”.

Page 3. Line 95:XRD diffraction patterns” has been revised as “X-ray diffraction (XRD) patterns”.

Page 3. Line 95:LSPR” has been revised as “local surface plasmon resonance (LSPR)”.

Page 3. Line 95: face-centered cubic” has been revised as “face-centered cubic (FCC)”.

Page 4. Line 164: “reducing agent” has been revised as “a reducing agent”.

Page 6. Line 200: “fabric” has been revised as “fabrics”.

Page 6. Line 204: “fabric with the different structures” has been revised as “fabrics with different structures”.

Page 6. Line 216: “tend” has been revised as “tends”.

Page 6. Line 219: “of” has been revised as “in”.

Page 7. Line 235-239:

Figure 3. (a) Large-pore polyester fabric before stretching.  (b) Large-pore polyester fabric after stretching.  (c) Small-pore polyester fabric before stretching. (d) Small-pore polyester fabric after stretching.  (e) Running-stitch polyester fabric before stretching. (f) Running-stitch polyester fabric after stretching. (g) Ribbed polyester fabric before stretching. (h) Ribbed polyester fabric after stretching.

Figure 4. (a) SEM image of conductive fabric prepared by large-pore polyester fabric, and the inset is the physical picture of conductive fabric prepared by large-pore polyester fabric. (b) Magnification of large pore polyester fabric surface. (c) SEM image of conductive fabric prepared by small-pore polyester fabric, the inset is the physical image of conductive fabric prepared by small-pore polyester fabric. (d) Magnification of small pore polyester fabric surface. (e) SEM image of conductive fabric prepared by running-stitch polyester fabric, the inset is the physical picture of conductive fabric prepared by running-stitch polyester fabric.  (f) Magnification of running-stitch polyester fabric surface. (g) SEM image of conductive fabric prepared by ribbed polyester fabric, and the inset is the physical image of conductive fabric prepared by ribbed polyester fabric. (h) Magnification of ribbed polyester fabric surface.

Page 7. Line 227: “Figure 3(c)(d) and (g)(h)” has been revised as “Figure 4(a)-(d)”.

Page 7. Line 229: “Figure 3(k)(l) and (o)(p)” has been revised as “Figure 4(e)-(h)”.

Page 7. Line 241: “Figure4ab” has been revised as “Figure 5ab”.

Page 7. Line 242: “physical” has been revised as “The physical”.

Page 7. Line 243: “Physical” has been revised as “The physical”.

Page 7. Line 243: “Figure 4(d) and (e)” has been revised as “Figure 5(d)(e)”.

Page 7. Line 245: “Figure 4(c) and (f)” has been revised as “Figure 5(c)(f)”.

Page 7. Line 249: “nanoparticles” has been revised as “nanoplates”.

Page 7. Line 249: “shows” has been revised as “show”.

Page 7. Line 250: “Figure 4(g)(h)” has been revised as “Figure 5(g)(h)”.

Page 7. Line 259: “Figure 5” has been revised as “Figure 6”.

Page 7. Line 260: “(e) and (f)” has been revised as “(e)(f)”.

Page 8. Line 260: “at high temperature” has been revised as “at a high temperature”.

Page 8. Line 263: “Figure 5(a) and (b)” has been revised as “Figure 6(a)(b)”.

Page 8. Line 267: “Figure 5(c)(d)” has been revised as “Figure 6(c)(d)”.

Page 8. Line 270: “Figure 5(e)(f)” has been revised as “Figure 6(e)(f)”.

Page 8. Line 282: “grows” has been revised as “grow”.

Page 9. Line 274: “Figure 6” has been revised as “Figure 7”.

Page 9. Line 287: “, and” has been revised as “,”.

Page 9. Line 290: “Van der Waals force” has been revised as “Van der Waals forces”.

Page 9. Line 298: “covers” has been revised as “cover”.

Page 10. Line 299: “Figure 7(a)” has been revised as “Figure 8(a)”.

Page 10. Line 306: “Figure 7(b)” has been revised as “Figure 8(b)”.

Page 10. Line 316: “Figure 7(c)” has been revised as “Figure 8(c)”.

Page 10. Line 333: “stretching for many times” has been revised as “stretching many times”.

Page 10. Line 325-326: “Figure 7(d)” has been revised as “Figure 8(d)”.

Page 10. Line 335: “releasing for 20 times” has been revised as “releasing 20 times”.

Page 10. Line 336: “of” has been revised as “in”.

Page 10. Line 337: “stretching for 60 times” has been revised as “stretching 60 times”.

Page 10. Line 352: “show” has been revised as “shows”.

Page 10. Line 357: “gradually, and” has been revised as “gradually and”.

Page 11. Line 335: “Figure 7(e)(f)” has been revised as “Figure 8(e)(f)”.

Thanks again for the reviewer’s precious comments and careful corrections.

Sincerely yours,

Qingwei Liao

Reviewer 2 Report

Dear Authors,

I found it difficult to understand  Fig 3. Are the images on the left (stretching experiments) without nanosilver, but the images on the right (SEM) with nanosilver? It should be described, possibly split into two figures.

In Fig. 7 you show a peak at 380 nm and mention in the text 382 nm.

You show the results of resistance measurements (Fig. 7), but don´t describe the experiments (how is the set-up, how do you contact the fibre?).

You mention up to 100 times of the stretching process, was this done manually as on the photographs?

For some readers it might be helpful to explain abbreviations (PVP, XRD,..)

With kind regards

Author Response

First, we would like to express our thanks to the reviewers for their instructive comments concerning our manuscript entitled ‘In-situ growth of nanosilver on fabric for flexible stretchable electrodes’ (ijms-1976254). We have studied the comments carefully, and then carefully revised the whole manuscript. The revisions/ explanation corresponding to the comment are shown in the following (Reviewers' comments are in italic font):

  1. I found it difficult to understand Fig 3. Are the images on the left (stretching experiments) without nanosilver, but the images on the right (SEM) with nanosilver? It should be described, possibly split into two figures.

A: According to the comments of the reviewers, we divided Figure 3 into two pictures and adjusted the placement of the pictures. The specific modifications are as follows:

Page 7. Line 235-239:

Figure 3. (a) Large-pore polyester fabric before stretching.  (b) Large-pore polyester fabric after stretching.  (c) Small-pore polyester fabric before stretching. (d) Small-pore polyester fabric after stretching.  (e) Running-stitch polyester fabric before stretching. (f) Running-stitch polyester fabric after stretching. (g) Ribbed polyester fabric before stretching. (h) Ribbed polyester fabric after stretching.

Figure 4. (a) SEM image of conductive fabric prepared by large-pore polyester fabric, and the inset is the physical picture of conductive fabric prepared by large-pore polyester fabric. (b) Magnification of large pore polyester fabric surface. (c) SEM image of conductive fabric prepared by small-pore polyester fabric, the inset is the physical image of conductive fabric prepared by small-pore polyester fabric. (d) Magnification of small pore polyester fabric surface. (e) SEM image of conductive fabric prepared by running-stitch polyester fabric, the inset is the physical picture of conductive fabric prepared by running-stitch polyester fabric.  (f) Magnification of running-stitch polyester fabric surface. (g) SEM image of conductive fabric prepared by ribbed polyester fabric, and the inset is the physical image of conductive fabric prepared by ribbed polyester fabric. (h) Magnification of ribbed polyester fabric surface.

  1. In Fig. 7 you show a peak at 380 nm and mention in the text 382 nm.

A: In the revised draft, we revised the “382 nm” to “380 nm”.

  1. You show the results of resistance measurements (Fig. 7), but don´t describe the experiments (how is the set-up, how do you contact the fibre?).

A: We carefully considered the comments of the reviewer, and we added relevant description of the operation for measuring fabric resistance.

The four-probe method is used for measurement. For the instrument equipped with a special four-probe head with spring, the distance of the four probe heads is equal, and the four leads are connected to the square resistance tester. When the probe is inserted into the fabric, the value of the square resistance tester is read. For the same fabric, after 10 measurements, the maximum and minimum values were removed, and the mean tail-cut value was calculated as the final result.

  1. You mention up to 100 times of the stretching process, was this done manually as on the photographs?

A: Yes, as shown in the picture, with the help of a ruler as a guide, we manually stretch the fabric repeatedly on the operating table for testing.

  1. For some readers it might be helpful to explain abbreviations (PVP, XRD,..)

A: According to the comments of the reviewers, we checked all the abbreviations in the articles and added some abbreviations that are helpful to the readers.

Page 3. Line 95: “PVP” has been revised as “Polyvinylpyrrolidone(PVP)”.

Page 3. Line 102: “AgNWs” has been revised as “Ag nanowires(AgNWs)”.

Page 3. Line 95: “SEM” has been revised as “Scanning Electron Microscope (SEM)”.

Page 3. Line 95:XRD diffraction patterns” has been revised as “X-ray diffraction (XRD) patterns”.

Page 3. Line 95:LSPR” has been revised as “local surface plasmon resonance (LSPR)”.

Page 3. Line 95: face-centered cubic” has been revised as “face-centered cubic (FCC)”.

In this revising chance, we revised the manuscript over and over again, besides all revisions shown above, there are other modifications, which are shown as below:

Page 2. Line 50: “have” has been revised as “has”.

Page 2. Line 65: “of” has been revised as “in”.

Page 2. Line 66: “uses a material” has been revised as “uses material”.

Page 2. Line 75: “a reducing agent” has been revised as “reducing agent”.

Page 2. Line 82: “under tensile state” has been revised as “under a tensile state”.

Page 2. Line 92: “have stronger tensile proper-ties” has been revised as “has stronger tensile properties”.

Page 3. Line 105: “the solution A” has been revised as “solution A”.

Page 4. Line 164: “reducing agent” has been revised as “a reducing agent”.

Page 6. Line 200: “fabric” has been revised as “fabrics”.

Page 6. Line 204: “fabric with the different structures” has been revised as “fabrics with different structures”.

Page 6. Line 216: “tend” has been revised as “tends”.

Page 6. Line 219: “of” has been revised as “in”.

Page 7. Line 227: “Figure 3(c)(d) and (g)(h)” has been revised as “Figure 4(a)-(d)”.

Page 7. Line 229: “Figure 3(k)(l) and (o)(p)” has been revised as “Figure 4(e)-(h)”.

Page 7. Line 241: “Figure4ab” has been revised as “Figure 5ab”.

Page 7. Line 242: “physical” has been revised as “The physical”.

Page 7. Line 243: “Physical” has been revised as “The physical”.

Page 7. Line 243: “Figure 4(d) and (e)” has been revised as “Figure 5(d)(e)”.

Page 7. Line 245: “Figure 4(c) and (f)” has been revised as “Figure 5(c)(f)”.

Page 7. Line 249: “nanoparticles” has been revised as “nanoplates”.

Page 7. Line 249: “shows” has been revised as “show”.

Page 7. Line 250: “Figure 4(g)(h)” has been revised as “Figure 5(g)(h)”.

Page 7. Line 259: “Figure 5” has been revised as “Figure 6”.

Page 7. Line 260: “(e) and (f)” has been revised as “(e)(f)”.

Page 8. Line 260: “at high temperature” has been revised as “at a high temperature”.

Page 8. Line 263: “Figure 5(a) and (b)” has been revised as “Figure 6(a)(b)”.

Page 8. Line 267: “Figure 5(c)(d)” has been revised as “Figure 6(c)(d)”.

Page 8. Line 270: “Figure 5(e)(f)” has been revised as “Figure 6(e)(f)”.

Page 8. Line 282: “grows” has been revised as “grow”.

Page 9. Line 274: “Figure 6” has been revised as “Figure 7”.

Page 9. Line 287: “, and” has been revised as “,”.

Page 9. Line 290: “Van der Waals force” has been revised as “Van der Waals forces”.

Page 9. Line 298: “covers” has been revised as “cover”.

Page 10. Line 299: “Figure 7(a)” has been revised as “Figure 8(a)”.

Page 10. Line 306: “Figure 7(b)” has been revised as “Figure 8(b)”.

Page 10. Line 316: “Figure 7(c)” has been revised as “Figure 8(c)”.

Page 10. Line 333: “stretching for many times” has been revised as “stretching many times”.

Page 10. Line 325-326: “Figure 7(d)” has been revised as “Figure 8(d)”.

Page 10. Line 335: “releasing for 20 times” has been revised as “releasing 20 times”.

Page 10. Line 336: “of” has been revised as “in”.

Page 10. Line 337: “stretching for 60 times” has been revised as “stretching 60 times”.

Page 10. Line 352: “show” has been revised as “shows”.

Page 10. Line 357: “gradually, and” has been revised as “gradually and”.

Page 11. Line 335: “Figure 7(e)(f)” has been revised as “Figure 8(e)(f)”.

Thanks again for the reviewer’s precious comments and careful corrections.

Sincerely yours,

Qingwei Liao
